# NRPS-like ATRR in Plant-Parasitic Nematodes Involved in Glycine Betaine Metabolism to Promote Parasitism

**DOI:** 10.3390/ijms25084275

**Published:** 2024-04-12

**Authors:** Hongxia Zhang, Yanlin Li, Jian Ling, Jianlong Zhao, Yan Li, Zhenchuan Mao, Xinyue Cheng, Bingyan Xie

**Affiliations:** 1College of Horticulture, Hunan Agricultural University, Changsha 410128, China; 2State Key Laboratory of Vegetable Biobreeding, Institute of Vegetables and Flower, Chinese Academy of Agricultural Sciences, Beijing 100081, China; 3College of Life Sciences, Beijing Normal University, Beijing 100875, China

**Keywords:** plant-parasitic nematode, *Meloidogyne incognita*, NRPS-like ATRR, glycine betaine reductase, host-derived RNA interference, plant–nematode interaction

## Abstract

Plant-parasitic nematodes (PPNs) are among the most serious phytopathogens and cause widespread and serious damage in major crops. In this study, using a genome mining method, we identified nonribosomal peptide synthetase (NRPS)-like enzymes in genomes of plant-parasitic nematodes, which are conserved with two consecutive reducing domains at the N-terminus (A-T-R_1_-R_2_) and homologous to fungal NRPS-like ATRR. We experimentally investigated the roles of the NRPS-like enzyme (MiATRR) in nematode (*Meloidogyne incognita*) parasitism. Heterologous expression of *Miatrr* in *Saccharomyces cerevisiae* can overcome the growth inhibition caused by high concentrations of glycine betaine. RT-qPCR detection shows that *Miatrr* is significantly upregulated at the early parasitic life stage (J2s in plants) of *M. incognita*. Host-derived *Miatrr* RNA interference (RNAi) in *Arabidopsis thaliana* can significantly decrease the number of galls and egg masses of *M. incognita*, as well as retard development and reduce the body size of the nematode. Although exogenous glycine betaine and choline have no obvious impact on the survival of free-living *M. incognita* J2s (pre-parasitic J2s), they impact the performance of the nematode in planta, especially in *Miatrr*-RNAi plants. Following application of exogenous glycine betaine and choline in the rhizosphere soil of *A. thaliana*, the numbers of galls and egg masses were obviously reduced by glycine betaine but increased by choline. Based on the knowledge about the function of fungal NRPS-like ATRR and the roles of glycine betaine in host plants and nematodes, we suggest that MiATRR is involved in nematode–plant interaction by acting as a glycine betaine reductase, converting glycine betaine to choline. This may be a universal strategy in plant-parasitic nematodes utilizing NRPS-like ATRR to promote their parasitism on host plants.

## 1. Introduction

Plant-parasitic nematodes (PPNs) threaten food security throughout the world. Recent years have also witnessed a worsening of the nematode problem [1]. During evolution, plant parasitism in the phylum Nematoda has arisen multiple times independently, resulting in a diversity of nematode parasitism strategies, such as serial dichotomy into ecto-parasites or endo-parasites, and migratory or sedentary parasites [1,2]. Moreover, PPNs appear with convergence in nematode physiology, such as stylet analogues, feeding tubes, etc., and plant physiology, such as feeding site analogues. Currently, over four thousand PPN species have been identified [3]. However, most economic losses are caused by a handful of sedentary PPN genera, especially root-knot nematodes, *Meloidogyne* spp., and cyst nematodes, *Heterodera* spp. and *Globodera* spp. [4]. These nematodes invade the roots of host plants and simultaneously modify plant physiology, development, metabolism, and immunity [5].

Plant–PPN interactions are dependent on sophisticated chemical signaling. In-depth understanding of metabolite-mediated plant–nematode interactions can guide us towards novel nematode management strategies [6]. Natural phytochemicals can interfere with parasitic nematode development. It was reported that application of marine brown alga (*Ascophyllum nodosum*) extracts to roots of *Arabidopsis thaliana* and tomato plants resulted in a significant reduction in the numbers of second-stage juveniles (J2s) and females of both *M. javanica* and *M. incognita* invading the roots, as well as the number of eggs recovered from the seaweed-extract-treated plants [7,8,9]. Extract of seaweed mainly contains betaines, including γ-aminobutyric acid betaine, δ-aminovaleric acid betaine, and glycine betaine [10]. Direct application of these betaines at concentrations equivalent to those in the extract also led to significant reductions in both the nematode invasion profile and egg recovery [8]. The molecular mechanisms of betaine toxicity are currently unknown. It was reported that betaine acts on a ligand-gated ion channel in the nervous system of the nematode *Caenorhabditis elegans* [11]. Among these betaines, glycine betaine is a ubiquitous non-canonical amino acid that acts as an osmolyte or as a methyl donor. It is a phytohormone-like plant growth and development regulator under stress conditions. It plays an essential role in the response of plants to abiotic stress [12]. Under abiotic stress conditions, with glycine betaine synthesized and accumulated or exogenously applied, many plant species (such as *Arabidopsis*, wheat, corn, barley, sorghum, soybean, alfalfa, tomato, pear, etc.) increase growth and development, such as plant height, leaf area, shoot biomass, stem length, root length and number, panicle length and weight, seed weight and number, as well as numbers of flowers and pods, and fruit size [12]. 

Accordingly, it is an interesting question how plant-parasitic nematodes adapt to glycine betaine in hosts. A recent study provides new thinking on this question. It was reported that a nonribosomal peptide synthetase (NRPS)-like glycine betaine reductase is highly conserved and widespread in the fungi kingdom. The NRPS-like enzyme has unusual domain architecture (ATRR), catalyzing two sequential two-electron reductions of glycine betaine to choline [13]. A genomic survey revealed that *atrr* genes are widespread but exclusive to eukaryotes, mostly in fungi, and also found in several protist and invertebrate species [13]. To investigate whether the coding gene of glycine betaine reductase exists in PPN genomes, we predicted secondary metabolite synthesis gene clusters containing NPRS in genomes of different nematodes, and mined NRPS-like genes from PPN genomes. However, it is unknown whether these PPN NRPS-like enzymes have the same function as fungal NRPS-like ATRR and whether they are associated with nematode parasitism. In view of that structure-guided approach to assign functions of uncharacterized multidomain proteins and potentially aid functional discovery of new enzymes [13], in this study, we first compared the protein structures of PPN NRPS-like enzymes and fungal NRPS-like. Then, we tested the roles of PPN NRPS-like genes on alleviating the toxicity of exogenous glycine betaine. Our results may provide a new perspective on PPNs utilizing host chemistry to promote parasitism.

## 2. Results

### 2.1. Discovery and Evolutionary Analysis of NRPS-like Genes in PPNs

For comparing secondary metabolite synthesis gene clusters containing NRPS in different nematode genomes, *NRPS* genes were mined from 62 representative nematode genomes (Appendix A) retrieved from WormBase (http://www.wormbase.org, release WS285, accessed on 14 May 2022) using antiSMASH 6.0 [14]. It was found that secondary metabolite synthetase genes in different nematode species vary greatly in numbers and types (Appendix A). In the clade I nematode genomes, no *NRPS* gene was found, while *NPRS* genes are abundant in most genomes of the clade III and IV nematodes. Notably, *NRPS-like* genes are universal in the genomes of the clade IV and parts of the clade III nematodes, but absent in the clade V nematode genomes (Figure 1A, Appendix A). To our surprise, in the genomes of root-knot and cyst nematodes, only *NRPS-like* genes were identified. Meanwhile, homologs of the gene encoding a hybrid polyketide-nonribosomal peptide (PKS-NRPS), which is the neuron-associated nematode synthase identified in *C. elegans* [15], are absent in these nematodes. 

Then, evolutionary analysis of 24 nematode NRPS-like proteins was performed. In view of the fact that the adenylation domain (A_domain) is the first domain the substrate encounters before it is added to the nascent peptide natural product [16], a phylogenetic tree inferred from amino-acid sequences of A_domain was constructed (Figure 1B). It is shown that the evolutionary relationship of nematode NRPS-like proteins is consistent with that of nematode species, i.e., nematodes in clades III and IV are obviously divided into two groups. Within the clade IV nematode group, nematodes with different feeding habits (plant-parasitic, animal-parasitic, and bacteria-feeding) are obviously separated and assigned to different clades. All PPNs are gathered to form a clade (Figure 1B). As the A_NRPS domain is substrate-specific, the orthologous relationships of PPN NRPS-like proteins indicate their similar functions.

Via multiple fragment cloning and sequencing, we obtained a full-length 3921 bp coding sequence (CDS) of the *NRPS-like* gene in *M. incognita*, encoding a protein composed of 1306 amino acid residues. Conserved domains of the protein were predicted by using NCBI CD-Search [17]. The result shows that the protein has four domains, i.e., adenylation (A) domain, thiolation (T) domain, and two successive reduction domains, R_1_ and R_2_, from the N-terminus to the C-terminus in order (Figure 1C). This domain structure (A-T-R_1_-R_2_) of the NPRS-like protein with an extra C-terminal YdfG-like short-chain dehydrogenase/reductase domain (R_2_) is similar to NRPS-like ATRR in fungi [13]. We accordingly named it MiATRR. 

### 2.2. Structural Similarity of PPN NRPS-like ATRR and Fungal NRPS-like ATRR

It has been reported that fungal NRPS-like ATRR is a glycine betaine reductase, which catalyzes reduction of glycine betaine to choline via the intermediate glycine betaine aldehyde, and is highly conserved and widespread in the fungi kingdom [13]. We compared the protein sequences of MiATRR and the fungal ATRR in *Aspergillus nidulans*. The similarity of the two proteins is 32.62%, indicating they are homologous. We then predicted the spatial structures of MiATRR and *A. nidulans* ATRR using RoseTTAFold server(https://robetta.bakerlab.org, accessed on 7 February 2024) [18] and compared their structures using TM-align [19]. The results show that the two proteins have similar spatial structures (Figure 2A). Although the TM-score value of the total protein is not high (0.3563), for each domain (A_domain, R_1_ and R_2_), the similarity between the two proteins is high (all TM-score value > 0.8) by separate comparison of each domain (Appendix A), indicating that the function of each domain is similar between the two proteins. 

It Is documented that the ATRR-A domain is substrate-specific and dictated by the amino acid residues lining the active-site pocket [20]. We then aligned the A_domain sequences of MiATRR orthologues in PPNs with *A. nidulans* ATRR and determined the active-site pocket of PPN ATRR (Figure 2B), based on the substrate-binding pocket (10 AA codes) of the glycine betaine reductase in fungi [13]. Among the 10 “pocket” sites, nine sites are conserved and one is variable (His/Ile) in PPNs. Compared with those of *A. nidulans* ATRR, five residue sites in PPN ATRRs are different from the fungal ATRR, but three of them are also variable among different fungal ATRRs [13]. The 3D structural alignment also shows the similar structure of the active-site pocket between MiATRR and *A. nidulans* ATRR (Figure 2C). It is widely accepted that the function of a protein can be determined if its structure is similar to other proteins whose functions are known. Therefore, we suppose that the function of MiATRR is similar to *A. nidulans* ATRR, which is a glycine betaine reductase for choline biosynthesis (Figure 2D).

### 2.3. Heterologous Miatrr Expression in Yeast Can Overcome the Growth Inhibition Caused by High Concentrations of Glycine Betaine

To investigate whether or not PPN ATRR can play a role as glycine betaine reductase, the full-length coding sequence of *Miatrr* was inserted into the vector pXK30 and then transformed into *Saccharomyces cerevisiae* BJ5464-NpgA, which is an engineered yeast strain containing a chromosomal copy of the 4′-phosphopantetheinyl transferase (PPTase) gene *npgA* from *Aspergillus nidulans* [21], for heterologous expression of *Miatrr*. Then, the yeast cells were spotted on YPEG media supplemented with different concentrations of glycine betaine (20 µM to 20 mM). The results of the dot plate test showed that the positive transformants can grow on media containing high concentration of glycine betaine (200 µM; 2 mM; 20 mM), which inhibited the growth of control strains (the wild-type strain and transformant with empty vector pXK30) (Figure 3). The results indicate that heterologous expression of *Miatrr* in *S. cerevisiae* BJ5464-NpgA can degrade glycine betaine in cells in a timely manner, suggesting that MiATRR is a glycine betaine reductase.

### 2.4. Upregulated Expression of Miatrr in the Early Infection Stage of M. incognita (Parasitic J2s)

To investigate the expression levels of *Miatrr* in different developmental stages of *M. incognita*, we first downloaded seven lifestage-specific transcriptome data of *M. incognita* from the INRA (http://www6.inra.fr/meloidogyne_incognita/Genomic-resources2/Downloads, accessed on 27 June 2022) (Appendix A). Then, we mapped the coding sequence of *Miatrr* against each transcriptome using BLASTn [22]. The results show that *Miatrr* is expressed in all life stages (including egg, free-living J2s, J2s in plant, J3–J4, and male), except in the female stage (Figure 4A, Appendix A).

Then, newly hatched free-living J2s of *M. incognita* were inoculated on seedlings of susceptible cucumber variety ZN6. Total RNA of the nematodes was extracted at different times after inoculation. The expression levels of *Miatrr* at different parasitic stages were detected via RT-qPCR. The results show that, compared with free-living J2s, the expression level of *Miatrr* obviously increased after inoculation. It reached a peak at 3 dpi (days post-inoculation), followed by that at 5 dpi (Figure 4B). We then checked the morphology of *M. incognita* in planta and found that the nematode was at parasitic J2 stage within 5 dpi (Figure 4C). The results indicate that *Miatrr* may play roles in the early stages of nematode parasitism.

### 2.5. Impacts of M. incognita Parasitism by Host-Derived Miatrr Silencing

For investigating the role of *Miatrr* in *M. incognita*, pSUPER-*Miatrr*-RNAi vector was constructed. Via *Agrobacterium tumefaciens*-mediated transformation, the silencing vector was transformed into *Arabidopsis thaliana*. After PCR confirmation (Appendix A), three transformed *A. thaliana* lines expressing the *Miatrr* hairpin double-stranded RNA (dsRNA) were used for *M. incognita* inoculation. After three days of infection on plant roots, the silencing effect on *Miatrr* by host-derived RNAi was detected. RT-qPCR assay showed that the expression level of *Miatrr* in *M. incognita* parasitizing in the *Miatrr*-RNAi lines was significantly reduced, in contrast to that of the nematode parasitizing in the two control lines, i.e., the wild-type plants and GFP RNAi line (Figure 5A). 

By examining the damage to plant roots after 42 days of *M. incognita* inoculation on different *A. thaliana* lines, we observed a significant decrease in the number of galls in *Miatrr*-RNAi lines (mean 18 galls/plant), compared with that in the WT (mean 44 galls/plant) and GFP-RNAi line (mean 43 galls/plant) (Figure 5B–D). We also dissected the galls and measured the size of the worms. It was found that the mean body width of worms in galls of the *Miatrr*-RNAi lines (mean 129.7–134.8 µm) was obviously smaller than that of worms in galls of the WT (mean 256.9 µm) and GFP-RNAi line (mean 258.5 µm) (Figure 5E–G). Also, the mean number of egg masses in the *Miatrr*-RNAi lines (mean 4/plant) was obviously smaller than that of the two controls (mean 9/plant) (Figure 5H). These differences of above three indices (mean numbers of galls and egg masses per plant, and mean nematode size) between the *Miatrr*-RNAi lines and the controls are significant. However, there were no obvious differences between the two controls (the wild-type plants and GFP-RNAi lines), nor among the three *Miatrr*-RNAi lines (Appendix A). The effects of RNAi *Miatrr* resulting in reductions in the numbers of galls and egg masses, as well as the size of worms, indicate that *Miatrr* plays an essential role in *M. incognita* parasitism.

### 2.6. Roles of Miatrr in Response to Exogenous Glycine Betaine and Choline

We first tested *M. incognita* tolerance to glycine betaine and choline in vitro. Free-living J2 worms were subjected to exposure in different concentrations of glycine betaine and choline. After 72 h of exposure, it was observed that free-living J2s showed good tolerance to both glycine betaine and choline. Whether exposed to low (20 μM) or high (200 μM, 2 mM, or 20 mM) concentrations, there was no obvious impact on the nematodes’ survival (Appendix A). Then, using RT-qPCR, we detected the expression levels of *Miatrr* in *M. incognita* after exposure to different concentrations of glycine betaine and choline for 72 h. The results show that neither glycine betaine nor choline can induce *Miatrr* expression change in *M. incognita* free-living J2s (Figure 6A, Appendix A).

We then tested the response of *M. incognita* to glycine betaine and choline in planta. After inoculation of the nematode on *A. thaliana* treated with a high concentration of exogenous glycine betaine (20 mM) for 72 h, the expression of *Miatrr* in J2s in plants (parasitic J2s) significantly increased, compared with that of parasitic J2s on the untreated plants. In contrast, following inoculation of the nematode on *A. thaliana* treated with a high concentration of exogenous choline (20 mM) for 72 h, the expression of *Miatrr* in parasitic J2s was significantly reduced, compared with that of parasitic J2s on the untreated plants (Figure 6B). The results indicate that the *Miatrr* expression of parasitic J2s is affected by glycine betaine and choline in the surrounding environment. 

We also observed the performance of *M. incognita* on WT and *Miatrr*-RNAi *A. thaliana* treated with exogenous glycine betaine or choline (20 mM). Galls and egg masses in roots were counted at 42 dpi. The results show that, for WT *A. thaliana*, the numbers of galls and egg masses on plants treated with glycine betaine (mean 36 galls/plant, 9 masses/plant) were obviously lower than those of plants treated with water (control, mean 40 galls/plant, 11 egg masses/plant) or treated with choline (43 galls/plant, 10 egg masses/plant) (Figure 6C). ANOVA analysis shows that the differences in galls (*df* = 2, 177, *F* = 7.955, *p* < 0.001) and egg masses (*df* = 2, 177, *F* = 3.316, *p* = 0.039) among the three treatments are significant at a statistical level. However, for *Miatrr*-Ri *A. thaliana*, the mean number of galls was only 5 per plant, and no egg mass was collected in plants treated with glycine betaine, which are significantly lower values than those of the control (mean 18 galls/plant, 5 egg masses/plant); meanwhile, in plants treated with choline, the numbers of galls and egg masses were more than the control (29 galls/plant, 8 egg masses/plant) (Figure 6D). The differences among the three treatments are statistically significant (for galls, *df* = 2, 117, *F* = 198.309, *p* < 0.001; for egg masses, *df* = 2, 177, *F* = 158.378, *p* < 0.001). Comparing the numbers of galls and egg masses in each treatment between WT and *Miatrr*-RNAi, all differences are statistically significant (*p* < 0.001, Appendix A). The above results show that exogenous glycine betaine can intensify the negative effects of RNAi *Miatrr*, i.e., the number of galls was further reduced and no egg mass was produced at 42 dpi, while exogenous choline can alleviate the negative effects of *Miatrr*-RNAi (Figure 6E,F), indicating that the parasitism of *Miatrr*-RNAi *M. incognita* can be significantly reduced by exogenous glycine betaine but be increased by exogenous choline. Therefore, it is suggested that the *Miatrr* gene in *M. incognita* is involved in glycine betaine and choline metabolism and related to nematode parasitism.

## 3. Discussion

NRPS and NRPS-like enzymes have different functions in primary and secondary metabolism, and they usually have significantly adaptive advantages for many organisms. Related studies have found that some NRPS enzymes play important roles in the growth and development of animals. In *C. elegans*, a hybrid polyketide–nonribosomal peptide produced by PKS-1 and NRPS-1 can promote recovery and survival during starvation-induced larval arrest [15]. An NRPS peptide produced by male schistosomes when paired with a female is responsible for inducing female sexual development and egg laying [23]. However, little is currently known about *NRPS* genes in plant-parasitic nematodes. In this study, via genome-mining-based discovery of NRPS genes from nematodes in different clades, we found that NRPS-like genes are mainly distributed in most of the clade IV and parts of the clade III nematode genomes, but not in the genomes of clade V nematodes (Figure 1A, Appendix A). Among the clade IV nematodes, *NRPS-like* genes are not found in the genomes of three mammal-parasitic nematodes (*Parastrongyloides trichosuri*, *Strongyloides stercoralis*, and *Steinernema scapterisci*). The phylogenetic relationships show that *NRPS-like* genes are correlated with the evolution and living habitat of nematodes (Figure 1B). All PPN *NRPS-like* genes are gathered within a clade. Bioinformatic analysis shows that they have conserved structure domains (A-T-R_1_-R_2_) (Figure 1C), which have high similarity with fungal NRPS-like ATRR (Appendix A). The A_domain of ATRR (ATRR-A), which selects a substrate and is the entry point to ATRR catalysis, plays a key role in peptide natural product biosynthesis [16]. The ATRR-A has a unique 10 AA code, and it is conserved among ATRR orthologs from different fungal species [13]. According to the alignment of ATRR-A sequences with *A. nidulans* ATRR, 10 “pocket” sites in PPN ATRR have been identified; nine of them are identical to amino acids in PPNs, and five are identical to *A. nidulans* ATRR. Proteins with high sequence identity and structural similarity tend to be characterized by functional conservation [24]. Based on then structure–function relationship, it is suggested that PPN ATRR is similar to fungal ATRR, functioning as a glycine betaine reductase that reduces glycine betaine to choline.

The function of PPN ATRR as a glycine betaine reductase was verified experimentally. Heterologous expression of *M. incognita Miatrr* in the yeast *S. cerevisiae*, which has no homolog of ATRR, can overcome the growth inhibition caused by high concentrations of glycine betaine (Figure 3). It is well known that nematode parasitism is conferred by the actions of a variety of genes that are upregulated during host infection. Via RT-qPCR detection, it was shown that the expression level of *Miatrr* is high in the parasitic J2 stage (at 3 dpi and 5 dpi) (Figure 4B), which is significantly higher than that of the free-living J2 (pre-J2), indicating that *Miatrr* is related to nematode parasitism. *Miatrr* expression is significantly knocked down by host-derived RNAi in *M. incognita*, and the nematode parasitism obviously decreases, with smaller numbers of galls and egg masses, as well as small body size (Figure 5). Comparing performances of the nematode on the wild-type and *Miatrr* RNAi *A. thaliana* plants treated with exogenous glycine betaine and choline (20 mM), it can be seen that the negative effects of RNAi *Miatrr* are intensified by glycine betaine, but alleviated by exogenous choline (Figure 6), indicating that *Miatrr* is involved in the glycine betaine and choline metabolism of the nematode and related to its parasitism.

Glycine betaine, an amphoteric quaternary amine, plays an important role as a compatible solute in plants under environmental stresses, such as high salt and low temperature [25]. It is a phytohormone-like plant growth and development regulator under stress conditions. Many crops (such as wheat, corn, barley, sorghum, soybean, alfalfa, tomato, etc.) under abiotic stress conditions synthesize and accumulate glycine betaine to increase growth and development [12]. Glycine betaine has been reported to act on a ligand-gated ion channel in the nervous system of the nematode *C. elegans* [11]. Alkaline extracts of the marine brown alga contain betaines, including γ-aminobutyric acid betaine, δ-aminovaleric acid betaine, and glycine betaine, which arrest larval development and suppress nematode fecundity [7,8,9]. However, in our study, glycine betaine had no obvious impact on the nematode *M. incognita* (Figure 6C, Appendix A). We suppose this is because the seaweed extract was a mixture of betaines, and the proportions of the other two betaines (γ-aminobutyric acid betaine, δ-aminovaleric acid betaine) were much higher than glycine betaine [8], so they may have contributed a main role in toxicity. Glycine betaine and choline may interconvert through redox reactions. In nematodes, choline is available for the synthesis of acetylcholine (Ach) by choline acetyltransferase (ChAT). Ach is the major excitatory transmitter at nematode neuromuscular junctions, and more than a third of the cells in the *C. elegans* nervous system release acetylcholine [26]. Cholinergic transmission is involved, directly or indirectly, in many *C. elegans* behaviors, including locomotion, egg laying, feeding, and mating. Therefore, choline is favorable to nematodes. Based on our current results, it is suggested that NRPS-like ATRR in PPN acts as a glycine betaine reductase, which utilizes plant metabolite glycine betaine and converts it to choline, which is needed by nematodes, thus promoting nematode adaptation on host plants. 

## 4. Materials and Methods 

### 4.1. Nematode and Plant Materials

Egg masses of *M. incognita* were collected from roots of infected *Ipomoea aquatica* and hatched in water. The free-living J2s were collected for RNA extraction and plant inoculation. Surface-sterilized seeds of *A. thaliana* (ecotype Columbia, Col-0) were sown on Murashige and Skoog (MS) medium (M519, PhytoTech, Lenexa, KS, USA) plates. After 10 days, the seedlings were transplanted into pots containing soil and vermiculite (3:1) and grown at 22 °C under photoperiod conditions of 16 h light/8 h dark. *Cucumis sativus* L. was grown at 28 °C under a 16 h/8 h photoperiod. 

### 4.2. Bioinformatics Analysis of NRPS-like ATRR in Nematodes

AntiSMASH 6.0 software [14] was used to mine NPRS gene clusters in the representative nematode genomes, which were retrieved from WormBase (https://wormbase.org/, accessed on 14 May 2022), with the parameter ‘Detection strictness’ relaxed. A list of the BioProject IDs of genomes is provided in Appendix A. We obtained the phylogenetic tree of nematode species from Lifemap (https://lifemap-ncbi.univ-lyon1.fr/, accessed on 3 September 2023) [27]. Conserved domains of NRPS-like proteins were predicted using NCBI CD-search [17]. For phylogenetic analysis of nematode NRPS-like proteins, we used NRPS-like A_domain sequences for alignment and tree building. The phylogenetic tree was built with MEGA X [28], using a maximum likelihood algorithm with bootstrap values 1000, LG + I + G model. The tree was visualized using iTOL [29]. 

We searched the 3D structure of *A. nidulans* ATRR (A0A1U8QWA2) in the SWISS-MODEL Repository [30] and downloaded the PDB format. The 3D structure of *M. incognita* MiATRR was predicted using the RoseTTAFold server (https://robetta.bakerlab.org, accessed on 7 February 2024). Protein structure comparison was performed with TM-align [19], using the TM-align server (https://zhanggroup.org/TM-align/, accessed on 9 February 2024) for comparative analysis. The TM-score of the comparative analysis was obtained through the TM-align method. To view the protein structure, we used PyMOL v2.5.4 (https://pymol.org/, accessed on 21 February 2024). 

RNA-seq transcriptome data for seven different developmental life-stages of *M. incognita* were retrieved from the INRA (http://www6.inra.fr/meloidogyne_incognita/Genomic-resources2/Downloads, accessed on 27 June 2022). To examine the expression of *Miatrr* in the different life stages’ transcriptome data (Appendix A), we aligned the coding sequence of MiATRR against the transcriptome data using BLASTn [22] with an E-value cutoff of 1 × 10^−5^.

### 4.3. Nematode Infection

*A. thaliana* seedlings (one month after transplant) were inoculated with free-living J2s. For nematode susceptibility assays, *A. thaliana* roots were inoculated with 200 free-living J2s per plant. The roots were collected, then galls and egg masses were counted under a dissecting microscope (Olympus, Tokyo, Japan) at 42 days post-inoculation (dpi).

### 4.4. RNA Isolation and Miatrr CDS Acquisition

Total RNA was extracted from newly hatched *M. incognita* free-living J2s or liquid-nitrogen (N_2_)-frozen plant roots with parasitic worms using Invitrogen™ TRIzol™ Reagent (15596026, Thermo Fisher, Waltham, MA, USA), according to the user guide. The complementary DNA synthesis and reverse transcription quantitative real-time PCR (RT-qPCR) were performed using a PrimeScript™ RT Reagent Kit with gDNA Eraser (RR047A; TaKaRa, Kyoto, Japan) according to the manufacturer’s instructions. The full-length coding sequences of *Miatrr* were obtained via multi-fragment cloning combined with fragment assembly. Three pairs of primers (Miatrr-s1-s3-F/-R) were designed for segmental amplification of *Miatrr* from cDNA, using Q5^®^ High-Fidelity DNA Polymerases (M0419, NEB, Beverly, MA, USA). The fragments were assembled using the ClonExpress MultiS One Step Cloning Kit (C113, Vazyme, Nanjing, China) with primer pair Miatrr-L/-R. Primer sequences are shown in Appendix A.

### 4.5. Reverse Transcription Quantitative Real-Time PCR (RT-qPCR)

For assay of expression levels of *Miatrr* in *M. incognita*, about 500 free-living J2s were inoculated per plant on *Cucumis sativus* seedlings (7 days hydroponic culture). Nematodes in roots were observed at 1, 3, 5, 7, 14, 21 dpi under a dissecting microscope. Total RNA was extracted each time, and RT-qPCR was performed on a Bio-Rad CFX96 (Bio-Rad, Hercules, CA, USA) real-time PCR system (all primers are shown in Appendix A) using the following amplification program: 95 °C for 5 min and 40 cycles of 95 °C for 30 s and 60 °C for 30 s. Data were processed using the 2^−ΔΔCT^ method [31]. The gene encoding glyceraldehyde-3-phosphate dehydrogenase (GAPDH) of *M. incognita* (*Minc12412*) was used as an internal control for the normalization of RT-qPCR data, as previously described [32]. At least three biological experiments with three technical replicates for each reaction were performed.

### 4.6. Heterologous Expression of MiATRR in Yeast

For heterologous expression in yeast, the full-length coding sequence of *Miatrr* was amplified with primer-pair pXK30-F/-R, and the corresponding PCR fragments were inserted into the linearized pXK30 vector (YEpADH2-Ura, yeast-*E. coli* shuttle vector) [33] to generate the pXK30::*Miatrr* construct. Then, the pXK30::*Miatrr* construct was transformed into competent *Escherichia coli* for cloning and sequence verification. The verified construct was transformed into *S. cerevisiae* BJ5464-NpgA using a Frozen-EZ Yeast Transformation II™ kit (T2001, Zymo Research, Irvine, CA, USA) according to the manufacturer’s instructions. Transformation with the empty vector pXK30 was used as control. The primers (synthesized by Tsingke Biotechnology, Beijing, China) used for cloning and plasmid construction are listed in Appendix A. Yeast strains harboring expression plasmids were grown in synthetic complete (SC) medium (2% dextrose, 0.2% amino acid mixture, 0.67% yeast nitrogen base, and 0.72 g/liter Ura DropOut supplement [Clontech]). Yeast cells were also cultured in YPD media (1% yeast extract, 2% Bacto-peptone, and 2% dextrose), or non-fermentable media (2% Bacto-peptone, 1% yeast extract, 2% ethanol, and 3% glycerol [Solarbio]). Solid media were prepared with the supplementation of 1.5% agar. Yeast cells were cultured at 30 °C.

For cell growth assays, cells cultured overnight in YPD media were diluted into fresh media (OD 600 = 0.2) and re-cultured to the mid-log phase (OD600 = 0.8–1.0). After dilution to OD600 = 0.1 and OD600 = 0.01 in sterilized water, 2 µL of cells were spotted on YPEG media supplemented with indicated concentrations of glycine betaine and choline (20 µM, 200 µM, 2 mM, 20 mM). Plates were incubated at 30 °C for 5 days before photography. Each assay was repeated three times, with two different transformants to confirm results. The wild-type yeast strain BJ5464-NpgA and the strain transformed with the empty pXK30 plasmid were grown on YPEG medium as positive controls. 

### 4.7. Generation of Transgenic A. thaliana Miatrr-RNAi Plants

For RNAi experiments in *A. thaliana*, a 277-nt *Miatrr* fragment was amplified from cDNA of *M. incognita* with two-pair primers using Q5^®^ High-Fidelity DNA Polymerases (M0419, NEB, MA, USA), and then inserted upstream and downstream of the pSAT5 intron in the forward and reverse orientations [34]. Then, after digestion with restriction enzymes *Xba*I and *Kpn*I, the hairpin fragment was acquired and inserted into the pSUPER destination vector to construct the pSUPER-*Miatrr*-RNAi vector. Via *Agrobacterium*-mediated transformation, the RNAi vector was then transformed to *A. thaliana* Col-0 (wild-type) according to the floral dip method, following the description in a previous study [35]. Lines were verified via PCR and semiquantitative RT-PCR after hygromycin screening. Homozygous T3 plants from three *Miatrr*-RNAi lines were used for RNAi effect assay. The homozygous GFP-RNAi T3 lines used as control were the same as described in previous study [36]. The primers (synthesized by Tsingke Biotechnology), restriction enzymes (NEB, Beverly, MA, USA), and T4 ligase enzyme (M0202, NEB, Beverly, MA, USA) used for plasmid construction are listed in Appendix A. 

### 4.8. Nematode Infection Assays

*A. thaliana* seedlings (one month after transplant) were used for nematode inoculation. About 200 free-living J2s of *M. incognita* were inoculated per plant of different *A. thaliana* lines (WT, GFP-RNAi and *Miatrr*-RNAi lines). At 42 days post-inoculation (dpi), the roots were collected. Galls and egg masses were counted, and width of body size of worms in galls was measured under a dissecting microscope (Olympus, Tokyo, Japan). Each treatment had 20 plants, and the experiment was repeated three times. For observation of nematodes in plant roots, roots were stained with acid fuchsin [37] and observed under a microscope (OLYMPUS IX53). Live nematodes appeared bright pink.

### 4.9. Test of M. incognita Tolerance to Glycine Betaine and Choline 

To test the tolerance of *M. incognita* to glycine betaine and choline, free-living J2s were subjected to different concentrations of chemicals. Glycine betaine and choline solutions were prepared with sterilized distilled water at gradient concentrations of 20 µM, 200 µM, 2 mM, and 20 mM. About 1000 free-living J2s of *M. incognita* were treated with the above solutions in 24-well plates. The plates were covered to prevent evaporation and maintained in the dark at 28 °C. After 72 h exposure to different concentrations of glycine betaine and choline solutions, the vitality (motile or paralyzed) of the worms was observed with the aid of an inverted microscope (OLYMPUS IX53, Olympus, Tokyo, Japan) at 20×, and percentage of survival was calculated. Treatment with no chemicals in distilled water was taken as control. Three replicate experiments were designed.

### 4.10. Impacts of Exogenous Glycine Betaine and Choline to M. incognita Parasitism

Seedlings (one month after transplant) of WT and *Miatrr*-RNAi *A. thaliana* were inoculated with 200 free-living J2s of *M. incognita*. Meanwhile, plants were watered with solution of exogenous glycine betaine or choline (20 mM) for one week (until 7 dpi); during this time the nematode was at parasitic J2 stage. Plants treated with clean water were taken as control. After 42 days of inoculation, the roots were collected. Galls and egg masses were counted under a dissecting microscope (Olympus, Tokyo, Japan). The weight of roots per WT plant treated with different chemicals was also measured, to evaluate the short-term effects of exogenous glycine betaine and choline on plant growth and biomass. 

### 4.11. Statistical Methods

The significance of the differences between groups was assessed using two-tailed *t*-tests. For multiple comparisons, we first performed Levene’s test for homogeneity of variance across groups and the Shapiro–Wilk normality test for Gaussian distribution. Then, ANOVA was conducted (F-test and Tukey test). A non-parametric test was also carried out via the Kruskal–Wallis method. The data were analyzed using IBM SPSS Statistics (Version 27, IBM, Chicago, IL, USA) and with GraphPad Prism v8.3.0 (https://www.graphpad.com/updates/prism-830-release-notes, accessed on 5 November 2022) for drawing figures.

## Figures and Tables

**Figure 1 ijms-25-04275-f001:**
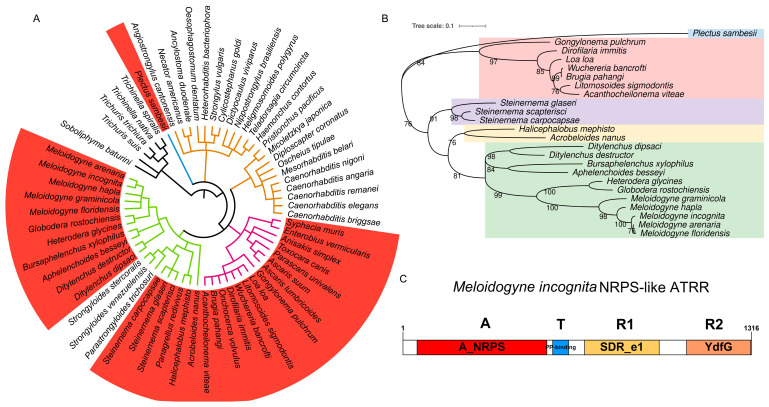
Identification of NRPS-like enzyme with A-T-R_1_-R_2_ domain structure in nematodes. (**A**) Distribution of NRPS-like in phylum Nematoda. Phylogenetic tree of nematode species is from Lifemap (https://lifemap-ncbi.univ-lyon1.fr/, accessed on 3 September 2023). Coral red ranges represent nematode species that possess *NRPS-like* genes. Branches with different colors represent different clades in Nematoda, i.e., black, clade I; blue, clade C; purple, clade III; green, clade IV; fulvous, clade V. (**B**) Phylogenetic relationship among nematode NRPS-like ATRR homologues. Phylogenetic tree generated in MEGA X based on NPRS_A domain sequences, using ML methods with bootstrap 1000. Different color ranges represent different nematode lifestyles. Hawkes blue, an aquatic nematode in clade C; pink, vertebrate parasitic nematodes in clade III; others are nematodes in clade IV: purple, insect-parasitic nematodes; yellow, free-living nematodes; light green, plant-parasitic nematodes. (**C**) Protein structural organization of MiATRR with four domains: A, A_NRPS, the adenylation domain of nonribosomal peptide synthetases (NRPS); T, PP-binding, phosphopantetheine attachment site; R1, SDR_e1, extended short-chain dehydrogenases/reductases subgroup 1; R2, YdfG, NADP-dependent 3-hydroxy acid dehydrogenase.

**Figure 2 ijms-25-04275-f002:**
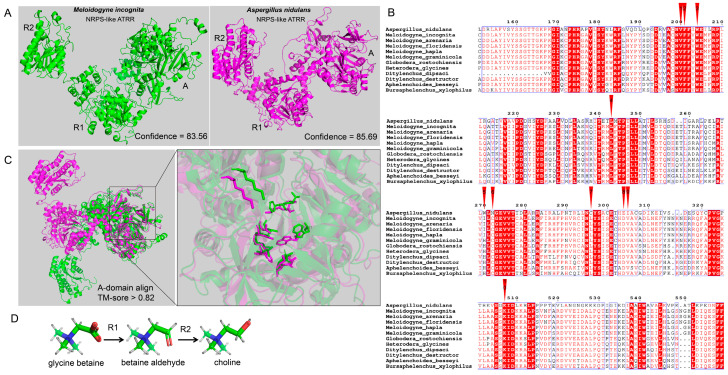
Structural comparison of PPN NRPS-like ATRR and fungal NRPS-like ATRR. (**A**) The 3D structures of MiATRR and fungal ATRR. The structure of MiATRR was predicted with RoseTTAFold online (job ID 570101), and the structure of *Aspergillus nidulans* ATRR (UniProt ID A0A1U8QWA2) was retrieved from AlphaFold Protein Structure Database. (**B**) Sequence alignment of ATRR-A from *A. nidulans* ATRR, MiATRR, and other PPN orthologues. The 10 “pocket” sites are indicated by red inverted triangles. (**C**) The 3D structural alignment of the active site pocket (10 residues) between MiATRR and *A. nidulans* ATRR. Green, MiATRR; pink, *A. nidulans* ATRR. (**D**) The two consecutive reducing domains of NRPS-like ATRR (R_1_, R_2_) reduce glycine betaine to choline via the intermediate glycine betaine aldehyde.

**Figure 3 ijms-25-04275-f003:**
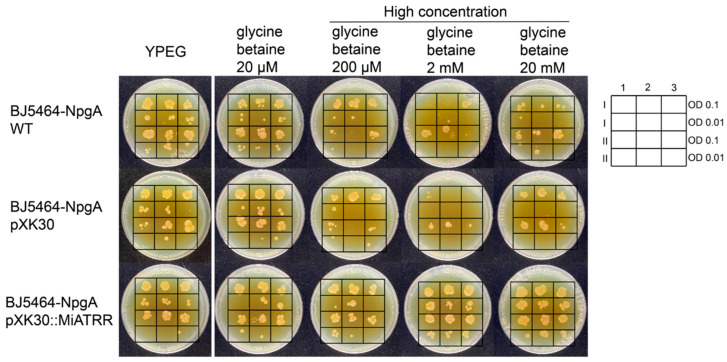
Growth of *Saccharomyces cerevisiae* BJ5464-NpgA transformed with *Miatrr* expression vector (pXK30::*Miatrr*) on YPEG medium supplemented with different concentrations of glycine betaine. The strain transformed with empty vector pXK30 and the wild-type strain were taken as controls. Overnight yeast cell suspensions were diluted and spotted on yeast peptone ethanol glycerol (YPEG) medium with 20 μM, 200 μM, 2 mM and 20 mM glycine betaine added, respectively. Medium without glycine betaine was taken as control. The lateral legend shows the arrangement of two spotted identification concentrations. Arabic numerals 1, 2, 3 represent three technical repeats, and Roman numerals I, II represent two biological repeats. The picture shows that heterologous expression of *Miatrr* in yeast can overcome the growth inhibition caused by high concentrations of glycine betaine.

**Figure 4 ijms-25-04275-f004:**
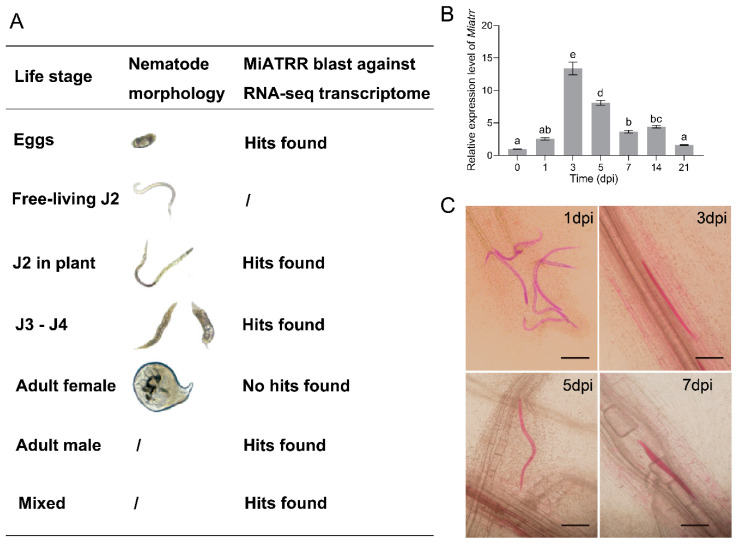
Expression of *Miatrr* in different life-stages of *M. incognita*. (**A**) Expression of *Miatrr* in lifestage-specific transcriptomes of *M. incognita*. Transcriptome data of *M. incognita* are from the 27 June 2022).(**B**) Expression profile of *Miatrr* at different times of *M. incognita* infection in plants (days post-inoculation, dpi). The relative expression level of *Miatrr* was quantified via RT-qPCR. The fold-change values were analyzed using the 2^−∆∆CT^ method. Means ± SE are shown. Different letters indicate significant differences based on Tukey HSD test (*p* < 0.05) (Appendix A). (**C**) Observation of *M. incognita* morphology in planta at different times. Roots were stained with acid fuchsin, and living nematodes appear bright pink. The bar represents 100 µm.

**Figure 5 ijms-25-04275-f005:**
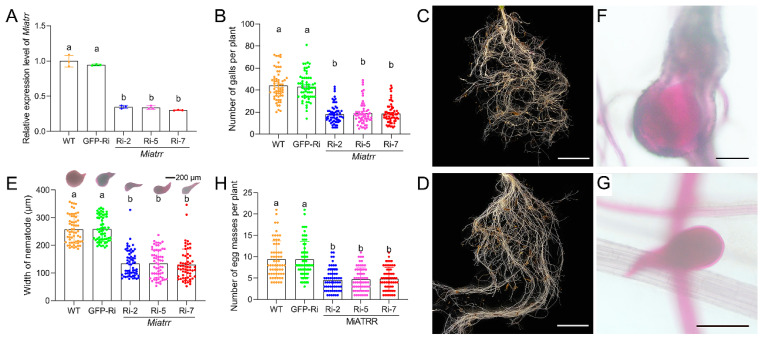
Effects of silencing *Miatrr* via host-derived RNA interference. (**A**) *Miatrr* expression levels in *M. incognita* inoculated on *Arabidopsis* plants of three RNAi lines (*Miatrr*-Ri-2, Miatrr-Ri-5, Miatrr-Ri-7), a *gfp*-RNAi line (GFP-Ri), and the wild type (WT) at 3 dpi, via RT-qPCR detection. The glyceraldehyde-3-phosphate dehydrogenase (GAPDH) coding gene was used as an internal control. The mean values are shown. Each bar represents the standard error (SE). Different letters indicate significant differences (*p* < 0.05). (**B**) The mean number of galls per plant at 42 dpi. Each bar represents the standard deviation (SD) in each *A. thaliana* line. Different letters indicate significant differences (*p* < 0.05). (**C**,**D**) Root galls on WT and Miatrr-Ri-2 plants, respectively. The bar represents 1 cm. (**E**) The mean width of worms at 42 dpi in roots of each *A. thaliana* line, measured after staining with acid fuchsin solution. Each bar represents SD. Different letters indicate significant differences (*p* < 0.05). (**F**,**G**) A common individual of *M. incognita* in galls of WT and Miatrr-Ri-2 plants, respectively. The bar represents 200 µm. Live nematode appears bright pink after staining with acid fuchsin solution. (**H**) The mean number of egg masses per plant of each *A. thaliana* line at 42 dpi. Each bar represents SD. Different letters indicate significant differences (*p* < 0.05).

**Figure 6 ijms-25-04275-f006:**
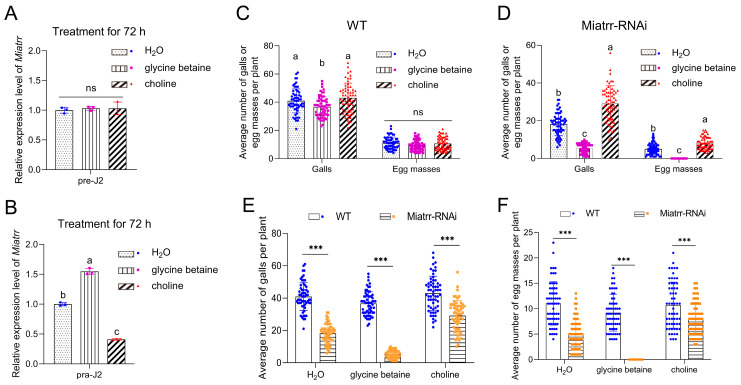
Impacts of exogenous glycine betaine and choline on *M. incognita* parasitism on *A. thaliana*. (**A**,**B**) Expression levels of *Miatrr* in free-living J2s (**A**) and J2s in plants (**B**) treated with exogenous glycine betaine and choline for 72 h, via RT-qPCR detection. The GAPDH gene was used as an internal control. Means ± SE are shown. Different letters indicate significant differences (*p* < 0.05). (**C**,**D**) The mean numbers of galls and egg masses of *M. incognita* per plant for WT (**C**) and *Miatrr*-RNAi *A. thaliana* (**D**) treated with exogenous glycine betaine or choline (20 mM) at 42 dpi. Each bar represents SD. Treatment with water was taken as control. Different letters indicate significant differences (*p* < 0.05), ‘ns’ represents no significant difference (*p* > 0.05). (**E**,**F**) Comparison of the mean number of galls (**E**) and egg masses (**F**) between WT and *Miatrr*-RNAi line of *A. thaliana* treated with exogenous glycine betaine or choline (20 mM) at 42 dpi. Each bar represents SD. *** indicates significant differences (*p* < 0.001, *t*-tests).

## Data Availability

Data is contained within the article and Appendix A.

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
