# Peer review of "NRPS-like ATRR in Plant-Parasitic Nematodes Involved in Glycine Betaine Metabolism to Promote Parasitism"

_ijms, 2024, doi:10.3390/ijms25084275_

Round 1

Reviewer 1 Report

Comments and Suggestions for Authors

1. "pre-J2s" and "par-J2s" seems not clearly presented the Juvenile status. I prefer to use "free-living J2s" and "J2s in plant" that should be more clear to reflect the J2 before and after invading plants.

2. Some sentences are too long to follow up, such as "Many plant species under abiotic stress 63 conditions, such as Arabidopsis, wheat, corn, barley, sorghum, soybean, alfalfa, tomato, 64 pear, etc., glycine betaine synthesized and accumulated or exogenously applied, increase 65 the plant growth and development, such as plant height, leaf area, shoot biomass, stem 66 length, root length, number and biomass, panicle length and weight, and seeds weight; as 67 well as the number of seeds, spikes, flowers and pods, and fruit size" in lines 65-68, others in lines 80-83, etc. It may be improved to separate the long sentences to short sentences.

3. Other concerns include "C" should be "Caenorhabditis" in line 60, "defect" should be "detect" in line 100, "lining" should be "?" in line 156, etc. Authors should carefully check the whole text avoid those errors

Reviewer 2 Report

Comments and Suggestions for Authors

This is an interesting manuscript concerning the analysis of the function of Meloidogyne incognita nonribosomal peptide synthetase (NRPS)-like enzyme for its parasitism. This is an important aspect of plant response to nematode attack and nematode evasion of plant defenses. This study addresses the economically important aspect. The research is well-planned, the methods are adequately selected and described. My major methodological concern is the use of a single reference gene for RT-qPCR analysis as it is currently considered insufficient. However, I am aware that multiple studies are still published that use single reference genes. Additionally, I am not sure that the authors have used appropriate statistical methods, I have included more information in comments. However, by looking at the graphics it seems and low p values it seems that even if the authors were forced to use a non-parametric Kruskal–Wallis test the differences should still be statistically significant. I do appreciate a reach of supplementary data. Although captions to tables S1-S3 could be added row data from figures 5 and 6 could be added as the next supplementary table. Concluding I recommend this manuscript to be accepted for publication in the section Molecular Plant Sciences of the International Journal of Molecular Sciences after minor revisions.

In-text comments:

Line 2: Please do not divide words in the title

Line 6: Please carefully check the affiliation formatting

Line 13: Please be more precise and add some statistics,

Line 13: Use the phrase “genome mining”

Line 17: consider rephrasing e.g. …enzyme in the nematode (Meloidogyne incognita (MiATRR)) parasitism.

Line 36:remove long -term

Line 65: Please correct this sentence

Line 67: increase the biomass, the biomass is uncountable

Line 76: to investigate

Line 101: Please upload a higher resolution version of this figure the names are hard to read> I suggest a vertical composition of subfigures, then the subfigures can be larger and easier to follow.

Line 102: Confirm that Figure S1A does not require copyright permission.

Line 183: The statement that this figure compares different strains is somewhat misleading, please rephrase, it is one strain transformed with an exogenous gene on the expression vector and an empty vector as a control. Additionally, it would be interesting to compare the activity of Aspergillus nidulans ATRR with MiTRR.

Line 217: Could you upload the row data from this experiment as a supplementary table.

Line 233: Please explain why you use one-way ANOVA and how you confirmed the assumptions, this information can be added to the supplementary data.

Line 235: p should be a lowercase

Line 250: Please use more precise language like two times, 20%, or significantly smaller etc.

Line 273: Please correct the labels of y axes

Line 283: Please correct the parasitic stage abbreviation sometimes it is “pra” and sometimes it is “par”

Line 306: result show

Line 328: habitat

Line 406: Provide access date

Line 436: Please note that single gene reference is currently considered a snot sufficient for RT-qPCR analysis.

Line 509: Please correct this data, it as it is different from what was described in the rest of the manuscript.

Comments on the Quality of English Language

 The language of this manuscript is understandable with no major issues, hover sometimes there are minor spelling mistakes, but I am afraid that I am not able to find them all. Sometimes the authors use some phrases such as obviously in places where I feel that they could be more precise.
